# Production of fatty acid-derived oleochemicals and biofuels by synthetic yeast cell factories

Yongjin J. Zhou[1,2], Nicolaas A. Buijs[1,†], Zhiwei Zhu[1,2], Jiufu Qin[1], Verena Siewers[1,2] & Jens Nielsen[1,2,3,4]

Sustainable production of oleochemicals requires establishment of cell factory platform strains. The yeast *Saccharomyces cerevisiae* is an attractive cell factory as new strains can be rapidly implemented into existing infrastructures such as bioethanol production plants. Here we show high-level production of free fatty acids (FFAs) in a yeast cell factory, and the production of alkanes and fatty alcohols from its descendants. The engineered strain produces up to $10.4\,g\,l^{-1}$ of FFAs, which is the highest reported titre to date. Furthermore, through screening of specific pathway enzymes, endogenous alcohol dehydrogenases and aldehyde reductases, we reconstruct efficient pathways for conversion of fatty acids to alkanes ($0.8\,mg\,l^{-1}$) and fatty alcohols ($1.5\,g\,l^{-1}$), to our knowledge the highest titres reported in *S. cerevisiae*. This should facilitate the construction of yeast cell factories for production of fatty acids derived products and even aldehyde-derived chemicals of high value.

[1] Department of Biology and Biological Engineering, Chalmers University of Technology, Kemivägen 10, Gothenburg SE-41296, Sweden. [2] Novo Nordisk Foundation Center for Biosustainability, Chalmers University of Technology, Gothenburg SE41296, Sweden. [3] Novo Nordisk Foundation Center for Biosustainability, Technical University of Denmark, Hørsholm DK2970, Denmark. [4] Science for Life Laboratory, Royal Institute of Technology, Stockholm SE-17121, Sweden. † Present address: Evolva Biotech, Lersø Parkallé 40-42, Copenhagen DK-2100, Denmark. Correspondence and requests for materials should be addressed to J.N. (email: nielsenj@chalmers.se).

Sustainable and cost-effective routes for renewable production of chemicals and fuels are needed to support the growing population and economy with a reduced carbon footprint[1,2]. Oleochemicals are substitutes of petrochemicals and are usually derived from plant oils and animal fats, which have limited availability[3]. Microbial fatty acid biosynthesis has captured much attention as it offers a way for renewable oleochemicals production[4]. There have been several reports on engineering the bacterium *Escherichia coli* for the production of various oleochemicals[5–11], including alkanes that can be used directly as biofuels[6]. On the other hand, for industrial scale production the yeast *Saccharomyces cerevisiae* is more suitable due to its robustness and tolerance towards harsh fermentation conditions, as well as its widespread use for bioethanol production[12,13]. This will allow transforming existing bioethanol production plants for production of these chemicals. The productivity and yield of oleochemicals produced by the well characterized model yeast *S. cerevisiae* is still relatively low[14–16]. Moreover, most biosynthetic pathways are designed to utilize the tightly regulated lipid biosynthesis intermediate fatty acyl-CoA[14] or fatty acyl carrier protein (ACP)[17], which limits the metabolic flux. Free fatty acids (FFAs) on the contrary can be accumulated to much higher levels ($>200$-fold higher than fatty acyl-CoA)[18] and used for the biosynthesis of alkanes and fatty alcohols through formation of a fatty aldehyde intermediate[7]. We thus explored the establishment of FFA-derived pathways for the production of alkanes and fatty alcohols, two classes of valued oleochemicals (Fig. 1).

We first constructed a plasmid-free yeast strain by blocking fatty acid activation and degradation, introducing an optimized acetyl-CoA pathway, expressing a more efficient fatty acid synthase (FAS) and overexpressing the endogenous acetyl-CoA carboxylase. The engineered strain produced up to $10.4\,g\,l^{-1}$ of FFAs in fed-batch fermentation. We then constructed biosynthetic pathways for production of alkanes and fatty alcohols by screening endogenous alcohol dehydrogenases/aldehyde reductases (ADH/ALRs) and pathway balancing, which resulted the highest titres of alkanes ($0.8\,mg\,l^{-1}$) and fatty alcohols ($1.5\,g\,l^{-1}$) in *S. cerevisiae*.

## Results

**Systematic engineering for free fatty acids production.** We first started by establishing a platform strain that overproduces FFAs. In *S. cerevisiae*, fatty acids are mainly synthesized *de novo* by a cytosolic type I FAS[19] as activated fatty acids (fatty acyl-CoAs) by condensing acetyl-CoA and malonyl-CoA. FFAs are rapidly re-activated by fatty acyl-CoA synthetases to fatty acyl-CoAs, whose accumulation feedback inhibits fatty acid biosynthesis[20]. A wild-type strain therefore only produced $3\,mg\,l^{-1}$ FFAs (Fig. 2a). To circumvent this, we interrupted the reactivation process by deleting two of the main fatty acyl-CoA synthetase encoding genes *FAA1* and *FAA4*. To prevent fatty acid degradation through β-oxidation we also deleted *POX1* encoding the fatty acyl-CoA oxidase, which catalyses the first step of this pathway. The resulting strain YJZ06 produced $0.56\,g\,l^{-1}$ FFAs (Fig. 2a). This is consistent with earlier studies, which have shown that interruption of FFA activation is essential for FFA accumulation and secretion[21,22]. Our previous[15] and current studies (*vide infra*) showed that deletion of the aldehyde dehydrogenase-encoding gene *HFD1* is essential for the production of fatty aldehyde-derived alkanes and fatty alcohols. Thus, we used the *HFD1* knockout strain YJZ08 for further engineering. To further increase FFA production we expressed a truncated *E. coli* thioesterase encoding gene *'tesA*

(refs 5,14) to increase FFA release from the FAS complex, which resulted in a titre of $0.67\,g\,l^{-1}$ (strain YJZ13).

Next we aimed on increasing the supply of the precursor cytosolic acetyl-CoA by introducing a synthetic chimeric citrate lyase pathway (Fig. 1), which has been proposed to play an important role in lipid accumulation in oleaginous yeasts[23]. In addition to expressing an ATP:citrate lyase (ACL) as described before[24], we here constructed and optimized the citrate lyase cycle (Figs 1 and 3a) by systematically comparing different heterologous ACLs and malic enzymes (MEs), two significant components of this pathway, and overexpressing the endogenous mitochondrial citrate transporter Ctp1 and malate dehydrogenase 'Mdh3. Introduction of the chimeric acetyl-CoA pathway, consisting of ACL and ME from *Rhodospuridium toruloides* combined with overexpression of Ctp1 and 'Mdh3, improved the growth of a pyruvate decarboxylase negative strain IMI076 with an internal deletion in MTH1 (Pdc − *MTH1-ΔT*)[25] (Fig. 3c). We further show that ACL from *Mus musculus* (MmACL) was better than the ones from *R. toruloides* (RtACL) and *Homo sapiens* (HsACL) in improving IMI076 growth (Fig. 3c) and ME from *R. toruloides* (RtME) was important for cell growth in addiction to ACL in IMI076 (Fig. 3d). Furthermore, the ACL-based acetyl-CoA pathway rescued the growth of pyruvate decarboxylase negative strain RWB837 (Fig. 3b), which is growth-deficient[25]. Consistently, plasmid-overexpression of these genes improved FFA production (Fig. 3e) and *MmACL* was better for FFA production compared with RtACL and HsACL (Fig. 3f). Since

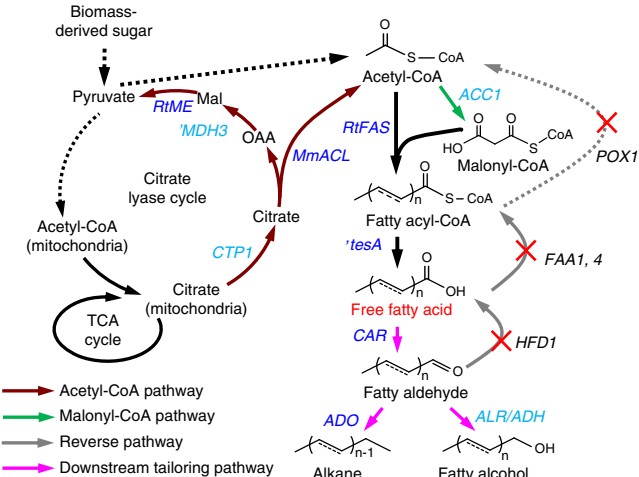

**Figure 1 | Establishing a yeast fatty acid platform for production of oleochemicals and biofuels.** The dotted lines indicate multiple steps and solid lines a single step. Overexpressed genes are shown in light blue (endogenous) or navy blue (heterologous). Reverse pathways were eliminated by deleting the corresponding genes (marked with X). For FFA production, fatty acyl-CoA synthetase encoding genes *FAA1* and *FAA4*, and fatty acyl-CoA oxidase encoding gene *POX1*, were disrupted. Furthermore, the truncated *E. coli* thioesterase 'TesA was overexpressed. For enhancing acetyl-CoA supply, a chimeric acetyl-CoA pathway, consisting of ACL (MmACL) from *Mus musculus*, ME (RtME) from *Rhodosporidium toruloides*, endogenous malate dehydrogenase with removed peroxisomal signal ('Mdh3) and citrate transporter Ctp1, was constructed and genome-integrated. For increased FFA biosynthesis, *R. toruloides* FAS encoding genes (*RtFAS1* and *RtFAS2*) were expressed through genome-integration and acetyl-CoA carboxylase encoding gene *ACC1* was overexpressed by promoter replacement. OAA, oxaloacetate; Mal, malate. For alkane/fatty alcohol production, a heterologous CAR from *Mycobacterium marinum* was introduced for reducing FFAs to fatty aldehydes, which were then transformed to alkanes by an ADO, or fatty alcohols by an ADH or an ALR.

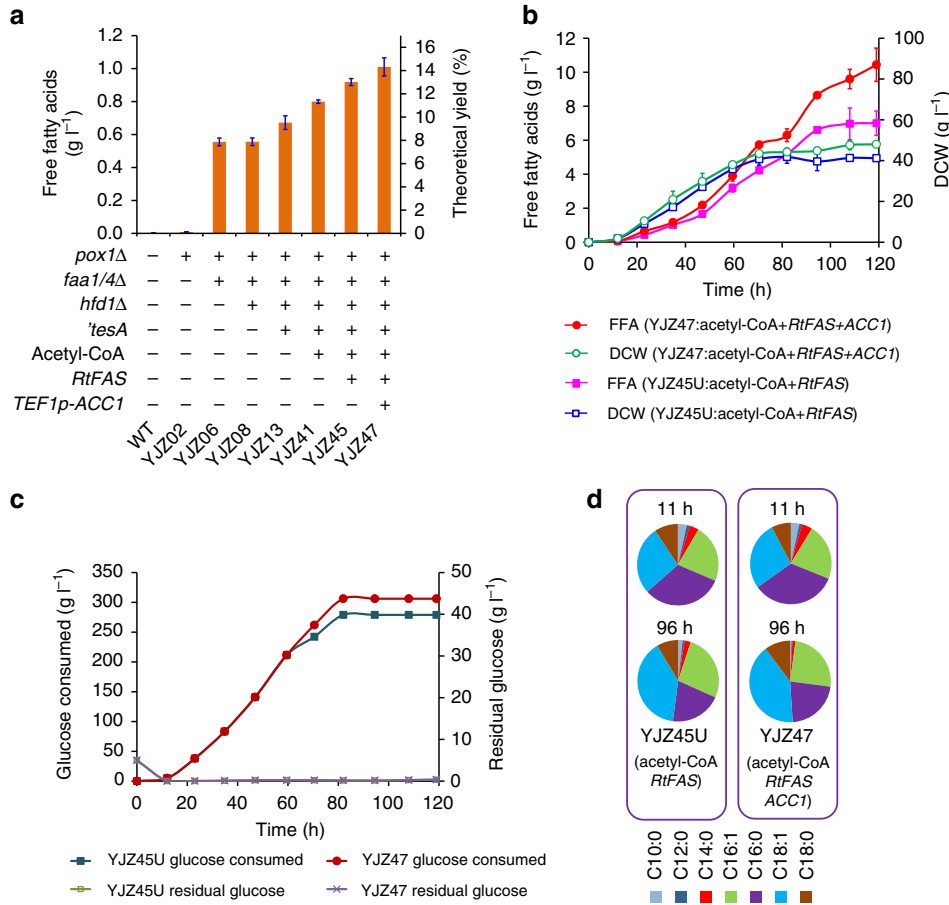

**Figure 2 | FFA production by engineered *S. cerevisiae* strains.** (**a**) FFA titres obtained with engineered strains in shake flasks after 72 h cultivation at 200 r.p.m., 30 °C. All data represent the mean ± s.d. of biological triplicates. (**b**) Fed-batch fermentation of strains YJZ45U and YJZ47. YJZ45U is a prototrophic strain with complementation of the *URA3* marker in YJZ45. Time courses of FFA titres (filled symbols) and cell mass (open symbols) are shown. (**c**) Glucose consumption profile and time courses of residual glucose during fed-batch fermentation. The data represent the mean ± s.d. of biological duplicates. (**d**) FFA profiles of the strain YJZ45 and YJZ47 at 11 h and 96 h.

plasmid-expression retarded the cell growth probably due to the metabolic burden (Supplementary Fig. 1), we thus genomic-integrated the optimized actyl-CoA pathway consisting of *MmACL, RtME, CTP1* and '*MDH3*, which improved FFA production to 0.80 g l$^{-1}$ (strain YJZ41, Fig. 2a).

Then we enhanced fatty acid synthesis by expressing a *R. toruloides* FAS (RtFAS). This FAS has two ACP domains, which may improve fatty acid biosynthesis efficiency by increasing the intermediate concentration in its reaction chamber[23,26]. *RtFAS* was functionally expressed and increased the total lipid and FFA content (Supplementary Fig. 2). Genomic integration of both *RtFAS* and the acetyl-CoA pathway (YJZ45) increased the FFA titre to 0.92 g l$^{-1}$ in shake flasks and the corresponding prototrophic strain YJZ45U reached 7.0 g l$^{-1}$ in fed-batch cultivation. After ensuring sufficient acetyl-CoA supply and fatty acid synthesis, we wanted to evaluate whether increased supply of malonyl-CoA, another tightly regulated precursor, could increase FFA production. We first evaluated an acetyl-CoA carboxylase mutant (Acc1$^{S1157A,S659A}$, Acc1**)[27] in which regulation by phosphorylation is abolished. However, its expression resulted in a lower FFA titre with lower biomass yield in fed-batch cultivation and promoted longer-chain fatty acid biosynthesis (Supplementary Fig. 3). The latter is consistent with a previous study reporting a shift towards C18 fatty acids at a higher malonyl-CoA/acetyl-CoA ratio by an *in vitro* reconstituted FAS from *S. cerevisiae*[28]. Alternatively, we

moderately enhanced the expression of the wild-type *ACC1* by replacing its native promoter with the *TEF1* promoter (strain YJZ47), which enabled an increase of FFA production to 1.0 g l$^{-1}$ (333-fold higher than wild-type strain, 14.3% of theoretical yield) in shake flask cultivation. It should be emphasized that the heavily engineered strain YJZ47 had a similar biomass yield compared with wild-type strain (Supplementary Fig. 4). This robustness is very important for implementation in industrial processes. Glucose limited fed-batch cultivation of this strain resulted in a titre of 10.4 g l$^{-1}$ FFAs (Fig. 2b,c), which was 49% higher than strain YJZ45U and also 20% higher than an engineered *E. coli* (8.6 g l$^{-1}$) in fed-batch culture[29] (Table 1). Interestingly, an increased percentage of oleic (C18:1) and stearic acid (C18:0) was observed in both strains during the fermentation (Fig. 2d and Supplementary Fig. 3c), which may be attributed to the upregulation of the fatty acid elongation system[30], since the yeast FAS has much higher level production of C16 fatty acids than C18 fatty acids *in vitro*[31].

**Engineering a fatty acid pathway for alkane production.** Subsequently, we wanted to exploit the FFAs for the production of alkanes, ideal drop-in biofuels[6]. We previously introduced a cyanobacterial fatty acyl-CoA-derived pathway, consisting of a *Synechococcus elongatus* fatty acyl-ACP/CoA reductase (AAR)

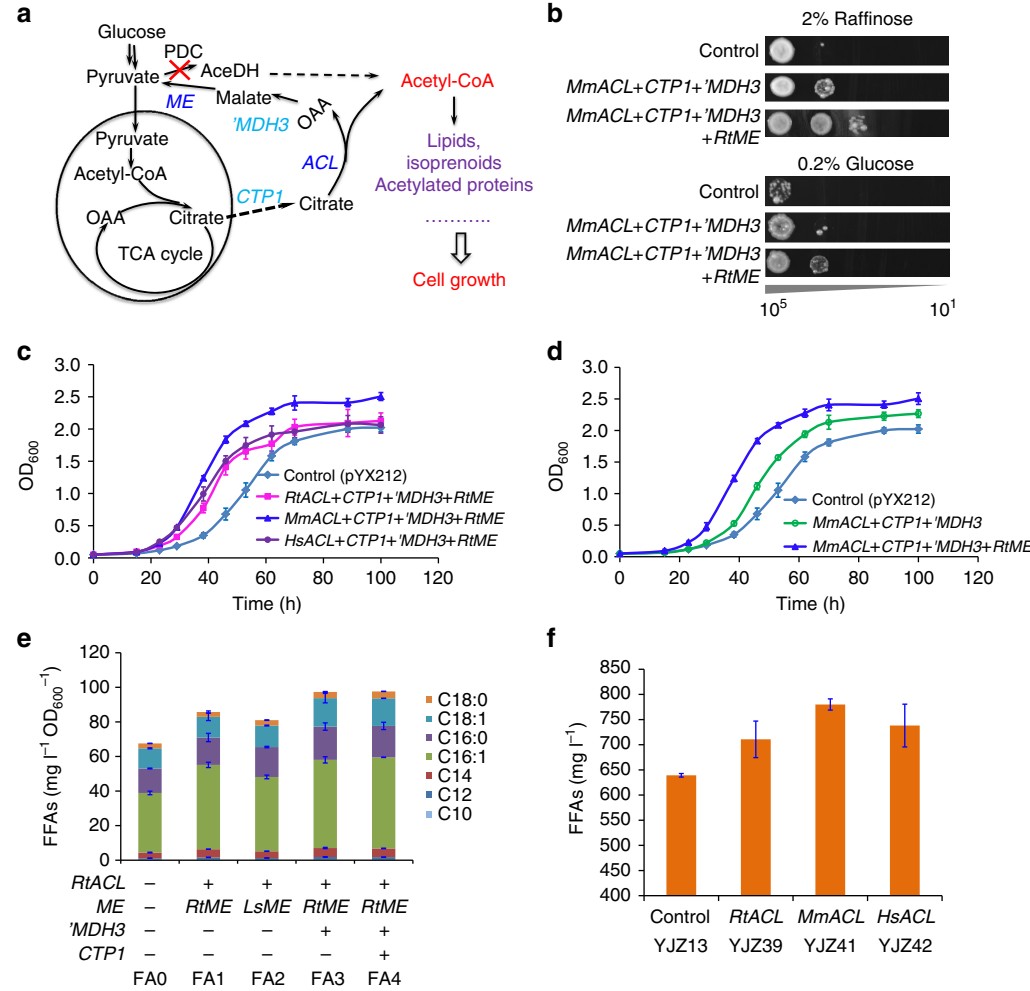

**Figure 3 | Optimization and characterization of ACL-based acetyl-CoA pathways in _S. cerevisiae_. (a)** Schematic illustration of the chimeric citrate lyase pathway for improved supply of acetyl-CoA, a key precursor for synthesis of cell building block. **(b)** The ACL-based acetyl-CoA pathway rescued the growth of PDC-negative strain RWB837, which is growth-deficient. **(c)** Introduction of the ACL-based acetyl-CoA pathway improved the growth of PDC-negative mutant strain _S. cerevisiae_ IMI076. Cells were cultured with an initial $OD_{600}$ of 0.05 in SC-URA at 30 °C, 200 r.p.m. **(d)** ME is beneficial for cell growth in addiction to ACL in the PDC-negative mutant strain _S. cerevisiae_ IMI076. **(e)** Introduction of the heterologous citrate lyase by-pass pathway improved FFA production. The engineered strains were constructed by transforming YJZ08 with the corresponding plasmids as shown in Supplementary Table 3. **(f)** Effect of different ACLs on production of FFAs. _RtACL_, _MmACL_ and _HsACL_ represent the optimized ACL genes from _R. toruloides_, _M. musculus_ and _H. sapiens_, respectively. The engineered strains were cultivated in shake flasks containing 15 ml minimal media for 72 h at 200 r.p.m., 30 °C. All data represent the mean ± s.d. of biological triplicates.

**Table 1 | Comparison of cell factories for production of free fatty acids.**

| Microorganism | Media | Cultivation mode | Titre (g l$^{-1}$) | Yield (% theoretical yield) | Reference |
|---|---|---|---|---|---|
| _E. coli_ | MM | Shake flask | 1.1 | 14 | 5 |
| _E. coli_ | SMM | Fed-batch | 8.6 | N.C.* | 29 |
| _E. coli_ | SMM | Fed-batch | 3.9 | N.C.* | 54 |
| _Y. lipolytica_ | MM | Shake flask | 0.5 | 7 | 43 |
| _S. cerevisiae_ | MM | Shake flask | 0.1-0.5 | 2-7 | 13,39,40 |
| _S. cerevisiae_ | YPD | Shake flask | 2.2 | N.C.* | 21 |
| _S. cerevisiae_ | MM | Shake flask | 1.0 | 14 | This study |
| _S. cerevisiae_ | MM | Fed-batch | 10.4 | 9 | This study |

MM, minimal media; SMM, semi-minimal media containing complex media component such as yeast extract; YPD, complex media containing 20 g l$^{-1}$ peptone, 10 g l$^{-1}$ yeast extract and 20 g l$^{-1}$ glucose.
*N.C.: not calculated due to containing complex media component such as yeast extract.

and fatty aldehyde deformylating oxygenase (_Se_ADO), in yeast and thereby demonstrated for the first-time production of alkanes in this organism[15]. The study, however, suggested that the AAR was inefficient in yeast. We therefore explored an alternative pathway by expressing a _Mycobacterium marinum_ carboxylic acid reductase (MmCAR)[7] (Fig. 4a). For activation of MmCAR[7], we

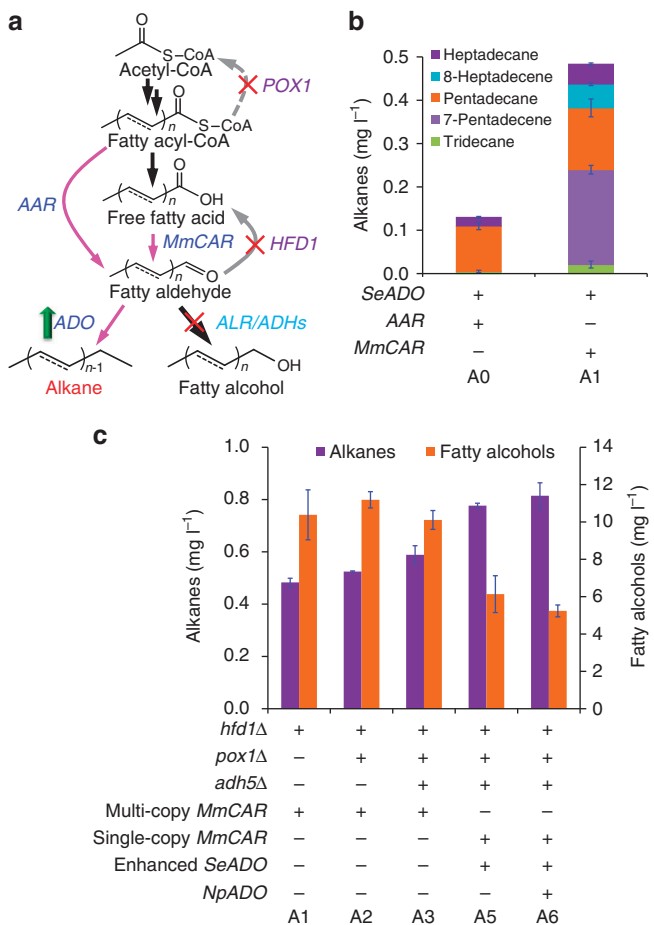

**Figure 4 | Alkane production from FFAs.** (**a**) The rewired metabolic pathways for enhancing alkane biosynthesis and decreasing accumulation of the by-product fatty alcohols. The alkane pathways are shown with pink arrows. (**b**) Alkane production by the FFA-derived (CAR and ADO) or the fatty acyl-CoA-derived (AAR and ADO) pathway. (**c**) Stepwise increasing alkane titres by eliminating competing pathways of aldehyde reduction and enhancing ADO expression, and corresponding fatty alcohol accumulation is also showed. The strains were cultivated in shake flasks for 72 h at 200 r.p.m., 30 °C. All data represent the mean ± s.d. of biological triplicates.

expressed 4′-phosphopantetheinyl transferase NpgA from *Aspergillus nidulans*. This FFA-based pathway enabled a 2.7-fold higher alkane production ($0.48 \, \mathrm{mg \, l^{-1}}$) than the fatty acyl-CoA-based pathway in an *hfd1Δ* background (Fig. 4b). Deletion of *POX1* slightly increased alkane production to $0.52 \, \mathrm{mg \, l^{-1}}$ (Fig. 4c). Further increasing the fatty acid supply did not increase the titre, but instead increased fatty alcohol production (Supplementary Fig. 6). Fatty alcohol accumulation might be caused by endogenous promiscuous aldehyde reductases (ALRs) and/or alcohol dehydrogenases (ADHs) that compete for the fatty aldehyde intermediates[32]. To solve this, we tried to identify the main competing enzymes by single deletion of 17 (putative) ALR/ADH-encoding genes (Supplementary Table 1). Of these, *ADH5* deletion led to an increased alkane production and decreased fatty alcohol accumulation (Fig. 4c and Supplementary Fig. 7). To further increase flux towards alkanes we increased the expression of the ADO by expressing *SeADO* under control of strong promoter UAS-TDH3p (ref. 33) and modulated *MmCAR* expression by single-copy genomic integration. The resulting strain A5 produced 50% more alkanes corresponding to $0.78 \, \mathrm{mg \, l^{-1}}$ and had a 40% reduction in fatty alcohol

accumulation, compared with the control strain A2 (Fig. 4c). Finally we evaluated additional expression of *Nostoc punctiforme NpADO* and this increased alkane production to $0.82 \, \mathrm{mg \, l^{-1}}$ with a further reduction in fatty alcohol accumulation (Fig. 4c). Although the titre is still cannot be comparable to *E. coli*, it represent more the eightfold higher titre than our previous work[15].

**Tailoring fatty acid for production of fatty alcohols.** The accumulation of fatty alcohols in our alkane producing strains (Fig. 3) gave us confidence to further explore the production of fatty alcohols from FFAs (Fig. 5a). Fatty alcohols are widely used as detergents, cosmetic ingredients and for the formulation of pharmaceuticals. Current fatty alcohol production strongly relies on plant oils, and microbial production could ensure a stable supply, without competition with food oil production, and enables tailored production of specific fatty alcohols. As observed for alkane production (Fig. 3b), the CAR was more efficient for fatty alcohol production than *Acinetobacter baylyi* fatty acyl-CoA/ACP reductase (ACR) or AAR (Supplementary Fig. 8). Since deletion of *ADH5* decreased fatty alcohol production in our ALR/ADH screening (Supplementary Fig. 7b,c), we overexpressed *ADH5* to increase fatty alcohol production. Indeed, *Adh5* was more efficient for fatty alcohol synthesis than several other ADH/ALRs, that is, endogenous Sfa1, Adh6, Adh7 or heterologous YjgB from *E. coli* (Supplementary Fig. 7d). When increasing the FFA supply (strain FOH6), the fatty alcohol production reached a titre of $23.2 \, \mathrm{mg \, l^{-1}}$ (Fig. 5b and Supplementary Fig. 9b). Allowing substrate channelling of the fatty aldehyde intermediates, by fusing MmCAR and Adh5, increased the fatty alcohol titre further by 26% (strain FOH21). However, enzyme fusion had a negative effect in the *HFD1* deletion strain FOH23 (Supplementary Fig. 9d), which may be attributed to the low activity of MmCAR in the fusion enzyme. Combining deletion of *HFD1* and blocking fatty acid degradation (strain FOH8) further increased fatty alcohol production to $61.2 \, \mathrm{mg \, l^{-1}}$ (Fig. 5b). However, there was still an accumulation of intracellular C18 fatty aldehydes (Supplementary Fig. 10b), indicating that C18 aldehyde reduction was a limiting step. Since a previous study showed that the bi-functional fatty acyl-CoA reductase *FaCoAR* from *Marinobacter aquaeolei* VT8 (ref. 34) has high activity towards long-chain fatty-aldehydes, we expressed *FaCoAR* instead of *ADH5* together with *MmCAR* in FOH28 and this resulted in $77.1 \, \mathrm{mg \, l^{-1}}$ fatty alcohols. Co-expression of *ADH5* and *FaCoAR* (strain FOH29) further improved fatty alcohol production to $81.8 \, \mathrm{mg \, l^{-1}}$ (Fig. 5b). Expression of *FaCoAR* and *ADH5* resulted in ∼80% reduction of the C18 fatty aldehyde (octadecanal and 9-octadecenal) content compared with *ADH5* overexpression (Supplementary Fig. 10b). We also evaluated fusion of *MmCAR* and *FacoAR*, but this decreased fatty alcohol production (Supplementary Fig. 10c). Our ADH/ALR knockout screening showed that *ADH6* deletion increases fatty alcohol production by 50% (Supplementary Fig. 7b). We therefore deleted *ADH6* (strain FOH31) resulting in increased fatty alcohol production to $89.5 \, \mathrm{mg \, l^{-1}}$ (Fig. 4b).

We found that there was still a high accumulation of FFAs (>2-fold higher than fatty alcohols, Supplementary Fig. 11) in strain FOH31, which indicated that fatty acid biosynthesis was overflown and the downstream reduction needed to be enhanced. We thus genome-integrated an additional copy of *MmCAR* under control of a *GAL7* promoter (together with *GAL80* deletion to enable gene expression without galactose addition). The resulting strain FOH33 produced 28% more fatty alcohols ($115 \, \mathrm{mg \, l^{-1}}$) with a 65% reduction in FFA accumulation (Fig. 5b and Supplementary Fig. 11). Glucose limited fed-batch cultivation

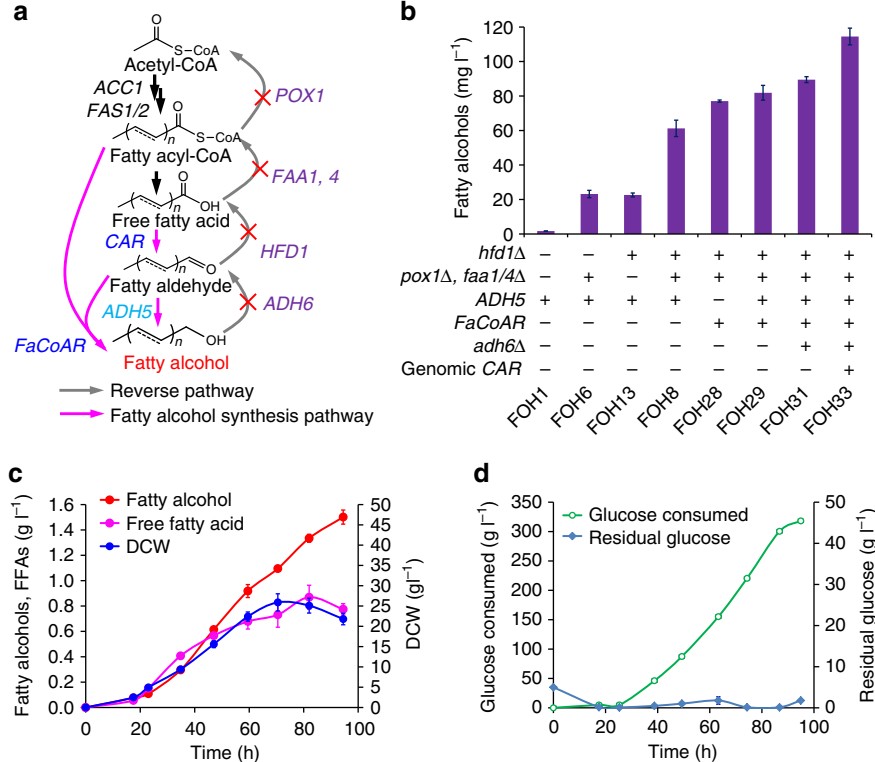

**Figure 5 | Engineering fatty alcohol production from FFAs. (a)** The rewired metabolic pathways for fatty alcohol production. Genes responsible for reverse reactions were deleted (marked with X), and genes related to fatty alcohol synthesis pathways were (over)-expressed. **(b)** Production of fatty alcohols in engineered strains in shake flasks, after 72 h cultivation at 200 r.p.m., 30 °C. All data represent the mean ± s.d. of biological triplicates. **(c)** Fed-batch fermentation of the best fatty alcohol producing strain FOH33 in 1 l bioreactor. **(d)** Glucose consumption profile and time courses of residual glucose during fed-batch fermentation. The data represent the mean ± s.d. of duplicates.

**Table 2 | Comparison of cell factories for production of fatty alcohols.**

| Microorganism | Media | Cultivation mode | Titre (g l$^{-1}$) | Yield (% theoretical Yield) | Reference |
|---|---|---|---|---|---|
| *E. coli* | MM | Fed-batch | 0.75 | 6 | 17 |
| *E. coli* | MM | Fed-batch | 1.75 | 8 | 55 |
| *E. coli* | MM | Fed-batch | 1.65 | 35 | 46 |
| *S. cerevisiae* | MM | Shake flask | 0.10 | 1.4 | 14,45 |
| *S. cerevisiae* | MM | Concentrated resting cells. Fed-batch | 1.11 | N.C.* | 35 |
| *S. cerevisiae* | MM | Shake flask | 0.12 | 1.7 | This study |
| *S. cerevisiae* | MM | Fed-batch | 1.51 | 1.4 | This study |

MM, minimal media.
*N.C.: not calculated due to the concentration of the cells with unknown initial fatty alcohols.

(Fig. 5d) of FOH33 had a more significant improvement (onefold, Supplementary Fig. 11c) in production of fatty alcohols (1.5 g l$^{-1}$, Fig. 5c), which is the highest reported titre of fatty alcohols produced by *S. cerevisiae* to date[14,35]. The titre is also comparable to *E. coli* cells though the yield is still much lower (Table 2).

## Discussion

The budding yeast *S. cerevisiae* is an attractive host for biosynthesis of specific products because of its robustness in industrial harsh conditions and easily transfer to existing bioethanol production plants. In this study, we undertook a major metabolic engineering effort to engineer *S. cerevisiae* for high-level production of FFAs and then their further transformation into alkanes and fatty alcohols. We demonstrated for the first time the significant conversion of FFAs to alkanes and fatty

alcohols in yeast, and we also showed that this FFA dependent pathway is far more efficient than the earlier reported route from fatty acyl-CoA (Fig. 3b and Supplementary Fig. 8). The production of alkanes and fatty alcohols benefited from our effort to streamline the fatty acid overproduction by taking the advantage of high cellular FFA levels ($>200$-fold higher than fatty acyl-CoA).

Oleaginous yeasts have been engineered for high-level production of neutral lipids such as triacylglycerol[36,37], an ideal feedstock for biodiesel production through transesterification. However, the intracellular accumulation requires very high cell density fermentation and also makes it challenging to recover the products[38]. FFAs are another ideal feedstocks for deoxygenated production of renewable hydrocarbon-based biofuels that are entirely fungible with fossil fuels[39]. More importantly, FFAs can be secreted (Supplementary Fig. 4c), which is beneficial for high-

level production by decoupling it from the cell growth (Fig. 2b). Aiming to overproduce FFAs, several researchers disrupted FFA activation and enhanced FFA biosynthesis, for example, through expression of different thioesterases, which enabled FFA production at 0.1–0.5 g l$^{-1}$ in minimal media in shake flask cultures (Table 1)[14,40,41]. More recently, disruption of FFA activation and neutral lipid recycle enabled production of 2.2 g l$^{-1}$ in complex (YPD) medium[21]. However, due to its high costs, complex makeup and variable composition, YPD medium would not be suitable for industrial production. Furthermore, the final engineered strain had a 20% lower biomass level in YPD medium, which indicated that the combination of disrupting FFA activation and neutral lipid recycle was harmful to the cell, and might retard growth further in minimal media with lower and less diverse nutrient availability. In this study, we systematically optimized the primary metabolism by disrupting FFA activation, constructing a more efficient fatty acid synthesis system and a chimeric citrate lyase cycle for enhanced precursor supply. More importantly, we are the first to construct a plasmid-free FFA overproducing strain by integration of all pathway components into the genome, which is important for application in industrial processes. These strategies enabled high-level FFA production in yeast under shake flask with minimal media (Fig. 2a) without a decrease in the biomass yield (Supplementary Fig. 4a). Fed-batch cultivation not only led to accumulation of a high FFA titre (10.4 g l$^{-1}$), but also a high biomass titre of 48 g l$^{-1}$, which is at the same level as a wild-type CEN.PK strain in fed-batch cultivation[42]. Before our study, the highest FFA titre (8.6 g l$^{-1}$) was reached by an engineered *E. coli* in fed-batch culture[29]. This is the first time that *S. cerevisiae* surpassed *E. coli* in regards to oleochemical production. It is worthy to mention that the FFA titre is also higher than oleaginous yeast *Yarrowia lipolytica* of 0.5 g l$^{-1}$ (Table 1)[43], which shows the potential of *S. cerevisiae* for FFA production.

Though lower in titre, the alkane production was much higher by using the FFA-based pathway compared with the fatty acyl-CoA-based pathway (Fig. 3b). By-product accumulation can hamper metabolic engineering endeavours. Because of the low ADO activity[44], the alkane titre remained low and fatty alcohols were being produced as major by-products (Fig. 3c). To overcome this problem, we first identified Adh5 as a key enzyme for conversion of fatty aldehydes to fatty alcohols by screening a series of ALR/ADH deletion strains. By deleting Adh5, we could significantly improve alkane production. However, their indispensable role in the biosynthesis of essential metabolites makes it impossible to delete all these enzymes. Increased expression of enzymes involved in conversion of fatty aldehydes to alkanes further increased alkane production, pointing to this step as having major flux control.

In contrast to alkane production, fatty alcohol biosynthesis relies on efficient reduction of fatty aldehyde (Fig. 5a). We therefore took advantage of our screening of different ALR/ADH deletion strains and found that overexpression of *ADH5* and deletion of *ADH6* could significantly improve fatty alcohol production (Supplementary Fig. 7). Combined with enhanced precursor supply, our final strain produced 1.5 g l$^{-1}$ fatty alcohols in fed-batch culture, which to our knowledge is the highest reported titre by *S. cerevisiae*. Current heterologous fatty alcohol biosynthesis pathways in yeast are designed to utilize fatty acyl-CoA as precursor, which enabled producing ~90 mg l$^{-1}$ fatty alcohols in shake flasks[14,45]. Recently, increasing acetyl-CoA supply and relieving the inhibition on fatty acyl-CoA biosynthesis, resulted in production of fatty alcohols at 330 mg l$^{-1}$ in shake flask and 1.1 g l$^{-1}$ in fed-batch cultivation with high concentrated cells[35]. In that study, concentrated cells were used in fed-batch cultivation, which might result in an

overestimated titre since concentrated cells should carry high-level initial cellular fatty alcohols. Moreover, the higher titre compared with our study for shake flask cultures might be attributed to the use of a dodecane overlay, which has been shown to be beneficial for fatty alcohol production[46]. However, a dodecane overlay will result in higher costs for product recovery due the similar boiling points of fatty alcohols and dodecane. Here, our strain produced much more fatty alcohols in fed-batch culture without a dodecane overlay. In the future, identification of fatty alcohol transporters might realize *in situ* product separation and recovery.

In conclusion, we have developed yeast cell factories for the production of FFAs and fatty alcohols, as well as demonstrated the significant production of alkanes in yeast. These strains represent a starting point for establishing yeast-based commercial bioprocesses for the production of oleochemicals and advanced biofuels from renewable resources. Our metabolic engineering strategies of pathway balancing at the fatty aldehyde node not only facilitated the production of fatty aldehyde-derived products but also provide valuable insights for construction of yeast cell factories for production of other valuable aldehyde chemicals, for example, vanillin[47], because of the similarity of the competition from ALR/ADHs.

## Methods

**Strains and reagents.** Plasmids and *S. cerevisiae* strains used are listed in Supplementary Tables 2 and 3. PrimeStar DNA polymerase was purchased from TaKaRa Bio. Taq DNA polymerase, restriction enzymes, DNA gel purification and plasmid extraction kits were purchased from Thermo Scientific. Yeast plasmid Miniprep I kits were purchased from Zymo Research. All oligonucleotides (Supplementary Table 4) were synthesized at Sigma-Aldrich. All chemicals including analytical standards were purchased from Sigma-Aldrich unless stated otherwise. All codon optimized heterologous genes were synthesized (Genscript) and listed in Supplementary Table 5.

**Strain cultivation.** Yeast strains for preparation of competent cells were cultivated in YPD consisting of 10 g l$^{-1}$ yeast extract (Merck Millipore), 20 g l$^{-1}$ peptone (Difco) and 20 g l$^{-1}$ glucose (Merck Millipore). Strains containing *URA3*-based plasmids or cassettes were selected on synthetic complete media without uracil (SC-URA), which consisted of 6.7 g l$^{-1}$ yeast nitrogen base (YNB) without amino acids (Formedium), 0.77 g l$^{-1}$ complete supplement mixture without uracil (CSM-URA, Formedium), 20 g l$^{-1}$ glucose (Merck Millipore) and 18 g l$^{-1}$ agar (Merck Millipore). The *URA3* maker was removed and selected against on SC + FOA plates, which contained 6.7 g l$^{-1}$ YNB, 0.77 g l$^{-1}$ complete supplement mixture and 0.8 g l$^{-1}$ 5-fluoroorotic acid. Strains containing the *kanMX* cassettes were selected on YPD plates containing 200 mg l$^{-1}$ G418 (Formedium).

Shake flask batch fermentations for production of alkanes and fatty alcohols were carried out in minimal medium containing 5 g l$^{-1}$ (NH$_4$)$_2$SO$_4$, 3 g l$^{-1}$ KH$_2$PO$_4$, 0.5 g l$^{-1}$ MgSO$_4$·7H$_2$O, 30 g l$^{-1}$ glucose, trace metal and vitamin solutions[48] supplemented with 40 mg l$^{-1}$ histidine and/or 60 mg l$^{-1}$ uracil if needed. While for production of FFAs, the minimal media was modified by using lower glucose (20 g l$^{-1}$) and higher KH$_2$PO$_4$ (14.4 g l$^{-1}$), which was beneficial for FFA accumulation (Supplementary Fig. 5). Cultures were inoculated, from 24 h precultures, at an initial OD$_{600}$ of 0.1 in 15 ml minimal medium and cultivated at 200 r.p.m., 30 °C for 72 h.

The batch and fed-batch fermentations for fatty acid and fatty alcohol production were performed in 1.0 l bioreactors, with an (initial) working volume of 0.4 l, in a DasGip Parallel Bioreactors System (DasGip). The initial batch fermentation was carried out in minimal medium containing 5 g l$^{-1}$ (NH$_4$)$_2$SO$_4$, 3 g l$^{-1}$ KH$_2$PO$_4$, 0.5 g l$^{-1}$ MgSO$_4$·7H$_2$O, 10 g l$^{-1}$ glucose, trace metal and vitamin solutions. The temperature, agitation, aeration and pH were monitored and controlled using a DasGip Control 4.0 System. The temperature was kept at 30 °C, initial agitation set to 600 r.p.m. and increased to maximally 1,200 r.p.m. depending on the dissolved oxygen level, aeration was provided at 30 sl h$^{-1}$ and the dissolved oxygen level was maintained above 40%, the pH was kept at 5.6 by automatic addition of 4 M KOH and 2 M HCl. The aeration was controlled and provided by a DasGip MX4/4 module. The composition of the off-gas was monitored using a Dasgip Offgas Analyzer GA4. Addition of the acid, base, and glucose feed was carried out with Dasgip MP8 multi-pump modules (pump head tubing: 0.5 mm ID, 1.0 mm wall thickness). The pumps, pH and DO probes were calibrated before the experiment. During the fed-batch cultivation, the cells were fed with an 800 g l$^{-1}$ glucose solution with a feed rate that was exponentially increasing ($\mu = 0.03$ h$^{-1}$) to maintain a constant biomass-specific glucose consumption rate. The initial feed rate was calculated using the biomass yield and concentration that were obtained

during prior duplicate batch cultivations with these strains. The feeding was started once the $CO_2$ levels dropped after the glucose was consumed.

Dry cell weight measurements were performed by filtrating 1 ml of broth through a weighed 0.45 μm filter membrane (Sartorius Biolab) and measuring the weight increase after drying for 48 h in a 65 °C oven. The filter was washed once before and three times after filtrating the broth with 5 ml deionized water.

**Genetic manipulation.** Seamless gene deletion was performed (Supplementary Fig. 12a) by using *Kluyveromyces lactis URA3* (*KlURA3*) as a selection marker, which was looped out by homologous recombination of the direct repeats, and selection on SC + FOA plates[49]. The deletion cassettes were constructed by fusing 200–600 nucleotide homologous arms with the *KlURA3*. For single gene deletion in identification of the ALRs and alcohol dehydrogenases, *kanMX* cassettes containing about 70 nucleotide homologous arms at both ends were used to transform strain YJZ03. *amdSYM* cassette[50] was used as a selection marker for genome-integration of FAS genes from *S. cerevisiae* (ScFAS) or *R. toruloides* (RtFAS). The pathways for alkane and alcohol production were assembled on a yeast chromosome or plasmid backbone pYX212 by using a modular pathway engineering strategy[51]. The gene expressing modules, consisting of a promoter, a structural gene, a terminator and the promoter of the next module for homologous recombination, were constructed by fusion PCR. Then the modules were gel purified and transformed to the *S. cerevisiae* with linearized plasmid pYX212.Genome-integration was performed by using a modular pathway integration strategy (Supplementary Fig. 12b). Taking the example of targeted integration of (TPIp-MmCAR-FBA1t) + (PGK1p-EcFNR-CYC1t) + (TEF1p-EcFD-TDH2t) + (tHXT7p-npgA) at the ADH5 locus in YJZ03, the whole pathway was divided into three modules of AK1, 2 and 3. In detail, the upstream homologous arm ADH5-up (from position − 382 to + 3) was amplified from CEN.PK113-11C genomic DNA with primer pair p59/p60. The AK1 module of ADH5-up + (TPIp-CAR-FBA1t) + CYC1t was assembled by fusing the parts of ADH5-up, TPIp-CAR-FBA1t + CYC1t. The part TPIp-CAR-FBA1t + CYC1t was amplified from the pAlkane16 by using primer pair p19/p31. The AK2 module of (CYC1t-EcFNR-PGK1p) + (TEF1p-EcFd-TDH2t) was amplified pAlkane16 by using the primer pair p32/p34. The AK3 module of TDH2t + (tHXT7p-npgA) + URA3 + ADH5-3' was assembled by fusing the DNA parts of TDH2t, tHXT7p-npgA, KlURA3 and ADH5-3'. The TDH2t was amplified from yeast genome DNA by using primer pair p15/p63. The tHXT7p-npgA was amplified from pAlkane16 with primer pair p27/p64. Amplification of KlURA3 was performed by using primer p65/p66 and pWJ1042 as a template. And downstream homologous arms ADH5-3' (from position + 579 to + 945) was amplified from CEN.PK113-11C genomic DNA by using primer pair p61/p62. Then the three modules (AK1, 2 and 3) were transformed into YJZ03 and transformants were selected on SC-URA plates (6.7 g l$^{-1}$ YNB without amino acids, 0.77 g l$^{-1}$ complete supplement mixture without uracil and 20 g l$^{-1}$ glucose and 15 g l$^{-1}$ agar). Clones were verified by colony PCR. Subsequently, 2–3 clones with correct module integration were cultivated overnight in YPD liquid medium and then plated on SC + FOA plates after wash for looping out of URA3 and also the 3' end of the ADH5 (from + 579 to + 1,056 that was left in place after the first round integration). All other pathways were integrated as above and the genetic arrangement is shown in Supplementary Fig. 12c

**Metabolite extraction and analysis.** FFAs were simultaneously extracted and methylated by dichloromethane containing methyl iodide as methyl donor[52]. Since the FFAs were secreted and cell culture formed an emulsion (Supplementary Fig. 4c), the cell culture should be mixed well before sample taking. Cell cultures from shake flask were diluted twofold with water and those from bioreactor were diluted 10-fold. Briefly, 200 μl aliquots of cell culture dilutions were taken into glass vials from 72 h incubated cultures, then 10 μl 40% tetrabutylammonium hydroxide (base catalyst) was added immediately followed by addition of 200 μl dichloromethane containing 200 mM methyl iodide as methyl donor and 100 mg l$^{-1}$ pentadecanoic acid as an internal standard. The mixtures were shaken for 30 min at 1,400 r.p.m. by using a vortex mixer, and then centrifuged at 5,000 g to promote phase separation. A 160 μl dichloromethane layer was transferred into a GC vial with glass insert, and evaporated 4 h to dryness. The extracted methyl esters were resuspended in 160 μl hexane and then analysed by gas chromatography (Focus GC, ThermoFisher Scientific) equipped with a Zebron ZB-5MS GUARDIAN capillary column (30 m × 0.25 mm × 0.25 μm, Phenomenex) and a DSQII mass spectrometer (ThermoFisher Scientific). The GC program was as follows: initial temperature of 40 °C, hold for 2 min; ramp to 130 °C at a rate of 30 °C per minute, then raised to 280 °C at a rate of 10 °C per min and hold for 3 min. The temperature of inlet, mass transfer line and ion source were kept at 280, 300 and 230 °C, respectively. The injection volume was 1 μl. The flow rate of the carrier gas (helium) was set to 1.0 ml min$^{-1}$, and data were acquired at full-scan mode (50–650 m/z). Final quantification was performed using the Xcalibur software.

For alkane and fatty alcohol quantification, cell pellets were collected from 5 ml (fatty alcohol) or 10 ml (alkane) cell culture and then freeze dried for 48 h. Metabolites were extracted by 2:1 chloroform:methanol solution[53], which contained hexadecane (alkanes) and pentadecanol (fatty alcohols) as internal standards. The extracted fraction was dried by rotary evaporation and dissolved in

hexane (alkanes) or ethyl acetate (fatty alcohols). Quantification of fatty alcohols and alkanes was performed on the same GC–MS system as used for fatty acid analysis. The GC program for alkane analysis was as follows: initial temperature of 50 °C, hold for 5 min; then ramp to 140 °C at a rate of 10 °C per min and hold for 10 min; ramp to 310 °C at a rate of 15 °C per min and hold for 7 min. The GC program for fatty alcohol quantification was as follow: initial temperature of 45 °C hold for 2.5 min; then ramp to 220 °C at a rate of 20 °C per min and hold for 2 min; ramp to 300 °C at a rate of 20 °C per min and hold for 5 min. The temperature of inlet, mass transfer line and ion source were kept at 250, 300 and 230 °C, respectively. The flow rate of the carrier gas (helium) was set at 1.0 ml min$^{-1}$, and data were acquired at full-scan mode (50–650 m/z). Final quantification was performed with Xcalibur software.

The extracellular glucose, ethanol and organic acid concentrations were determined by high-performance liquid chromatography analysis. To that end, a 1 ml broth sample was filtered through a 0.2 μm syringe filter and analysed on an Aminex HPX-87G column (Bio-Rad) on an Ultimate 3000 HPLC (Dionex Softron GmbH). The column was eluted with 5 mM $H_2SO_4$ at a flow rate of 0.6 ml min$^{-1}$ at 45 °C for 26 min.

**Data availability.** The authors declare that all data supporting the findings of this study are available within the article and its Supplementary Information file or available from the corresponding author upon reasonable request.

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

## Acknowledgements

This work was funded by Knut and Alice Wallenberg Foundation, Novo Nordisk Foundation, Vetenskapsrådet and FORMAS. This work is part of a collaborative project between Chalmers and Total. We thank Prof. Zongbao K. Zhao (Dalian institute of Chemical Physics, CAS, China) for kindly sharing the RtFAS encoding cDNA (*RtFAS1* and *RtFAS2*), Michael Gossing for help with constructing the *ACC1\*\** expression cassette, and Sakda Khoomrung and Julia Karlsson for their help with GC analysis. We also appreciate the helpful discussion with Mingtao Huang, Boyang Ji, Yun Chen, Paulo Teixeira, Paul Hudson (KTH, Stockholm, Sweden), Rahul Kumar and Guodong Liu.

## Author contributions

Y.J.Z. and J.N. conceived the study; Y.J.Z. designed and performed all the experiments and analysed the data; N.A.B. assisted with experimental design, data analysis and bioreactor studies; J.Q. assisted with bioreactor studies; Z.Z. assisted with constructing the single gene deletion strains of ADHs/ALRs and verifying the RtFAS function; Y.J.Z., N.A.B., Z.Z., V.S. and J.N. wrote the manuscript.

## Additional information

**Competing financial interests:** Y.J.Z., N.A.B., V.S. and J.N. have filed a patent (Engineering of hydrocarbon metabolism in yeast, No. WO2015057155 A1) for protection of part of the work described herein. All other authors declare no competing financial interests.

**How to cite this article**: Zhou, Y. J. *et al.* Production of fatty acid-derived oleochemicals and biofuels by synthetic yeast cell factories. *Nat. Commun.* 7:11709 doi: 10.1038/ncomms11709 (2016).

