## [Peer Review File · Nature Communications]

Reviewer #1

A&E The manuscript by Zhou et al. describes production of fatty acids, fatty alcohols, and alkanes in *Saccharomyces cerevisiae*. First they use various previously developed modifications to improve free fatty acid production in *S. cerevisiae*. Additionally, they install a pathway to produce acetyl-CoA from citrate, which is an important pathway to increase acetyl-CoA supply in oleaginous yeasts. The best strain produces 10.8 g/L free fatty acids in 120 hr under fed-batch fermentation conditions. This titer is very impressive. A glucose consumption profile should be added to Fig. 2 so that readers can estimate the yields during the production. Next, they extend the pathway to alkanes. The titers are very low (~1 mg/L) due to low activity of ADO, which catalyzes the conversion of fatty aldehydes to alkanes. Finally, they extend the pathway to fatty alcohols. They install two types of fatty alcohol production pathways (1. FaCoAR, 2. CAR-ADH) and remove several competing pathways. The best strain produces 1.5 g/L fatty alcohols under fed-batch fermentation conditions. For Fig. 5, a glucose consumption profile should be added. The fed-batch fermentation experiment in Fig. 2b produces 10 g/L free fatty acids. However, the fed-batch fermentation experiment in Fig. 5c produces only 1.5 g/L fatty alcohols and 0.8 g/L free fatty acids (total 2.3 g/L). Please comment on why the free fatty acid (+ fatty alcohol) production in the fatty alcohol strain is much lower than that in the free fatty acid strain.

B. Overall, the study is well-planned, logically presented, and provides sufficiently novel and interesting results. The manuscript should be appreciated among the metabolic engineering community.

C. adequate

D. adequate

F.

Line 27, 215, 217, & 241 Claims of "highest" or "first" should in general be deleted. The authors have no way of knowing what is or is not reported in conferences, in press, or in patents under examination.

Line 74 Since it is already known that this pathway plays an important role in increasing acetyl-CoA for lipid accumulation in oleaginous yeasts the pathway should not be described as a novel pathway.

Line 115 Please expand the explanation on why this is interesting.

G. adequate

H. well written

Reviewer #2

In this study, the authors engineered *S. cerevisiae* to overproduce free fatty acids and then transformed into alkanes and fatty alcohols via different metabolic engineering strategies. Through screening of specific pathway enzymes and endogenous alcohol dehydrogenases/aldehyde reductases, the authors obtained the highest titer of fatty acids, alkanes and fatty alcohols in *S. cerevisiae*. This is a significant milestone and breakthrough in fatty acid overproduction works in last five years. The quality of data and presentation is very high in this manuscript. The manuscript also well summarized the previous work and the conclusion is robustness. Overall, this paper is easy to read and worth publishing in Nature Communication.

Minor suggestions:

1. In Table 1, the authors compared cell factories for free fatty acids production, the authors could

also summarized cell factories for alkanes and fatty alcohols production. The authors should introduce and cite the following previous achievements by using *E. coli* in the introduction.

1. Quantitative analysis and engineering of fatty acid biosynthesis in *E. coli*.

Liu T, Vora H, Khosla C.

Metab Eng. 2010 Jul;12(4):378-86. doi: 10.1016/j.ymben.2010.02.003.

2. In vitro reconstitution and steady-state analysis of the fatty acid synthase from *Escherichia coli*.

Yu X, Liu T, Zhu F, Khosla C.

Proc Natl Acad Sci U S A. 2011 Nov 15;108(46):18643-8. doi:10.1073/pnas.1110852108.

3. Metabolic engineering of fatty acyl-ACP reductase-dependent pathway to improve fatty alcohol production in *Escherichia coli*.

Liu R, Zhu F, Lu L, Fu A, Lu J, Deng Z, Liu T.

Metab Eng. 2014 Mar;22:10-21. doi: 10.1016/j.ymben.2013.12.004.

2. When single deleted 17 putative ALR/ADH encoding genes, the result indicated that in addition to ADH6, ARA1 and NRE1 could also improve the level of fatty alcohol production, so why did the authors delete these genes at the same time, or did these genes inhibit the cell growth?

3. There are little mistakes in the manuscript. Examples: in line 81, " improved the can improve growth "; in Supplementary Fig. 3, "acetyl-CoA+tesA", " acetyl-CoA'tesA ", etc.

Response to reviewer comments

Dear Editor and Reviewers

Thanks for all your constructive comments and suggestions. Accordingly, we have modified our manuscript, and the changes are shown in red in the revised text. Here below is our point to point response to the reviewers' comments.

Responses to Reviewer 1

Question/Comment 1:

A&E The manuscript by Zhou et al. describes production of fatty acids, fatty alcohols, and alkanes in *Saccharomyces cerevisiae*. First they use various previously developed modifications to improve free fatty acid production in *S. cerevisiae*. Additionally, they install a pathway to produce acetyl-CoA from citrate, which is an important pathway to increase acetyl-CoA supply in oleaginous yeasts. The best strain produces 10.8 g/L free fatty acids in 120 hr under fed-batch fermentation conditions. This titer is very impressive. A glucose consumption profile should be added to Fig. 2 so that readers can estimate the yields during the production. Next, they extend the pathway to alkanes. The titers are very low (~1 mg/L) due to low activity of ADO, which catalyzes the conversion of fatty aldehydes to alkanes. Finally, they extend the pathway to fatty alcohols. They install two types of fatty alcohol production pathways (1. FaCoAR, 2. CAR-ADH) and remove several competing pathways. The best strain produces 1.5 g/L fatty alcohols under fed-batch fermentation conditions. For Fig. 5, a glucose consumption profile should be added. The fed-batch fermentation experiment in Fig. 2b produces 10 g/L free fatty acids. However, the fed-batch fermentation experiment in Fig. 5c produces only 1.5 g/L fatty alcohols and 0.8 g/L free fatty acids (total 2.3 g/L). Please comment on why the free fatty acid (+ fatty alcohol) production in the fatty alcohol strain is much lower than that in the free fatty acid strain.

Reply 1:

We thank the reviewer for the positive comments.

- 1) The glucose consumption profile for the fed batch experiment was added in Fig 2c and Fig 5d.
- 2) Concerning the comment about the lower titer of free fatty acid + fatty alcohol than that in the free fatty acid strain: the Free fatty acid production strain YJZ47 carries a genome integrated chimeric ACL pathway, RtFAS and overexpressed ACC1 with the deletion of HFd1, FAA1/4 and POX1, while fatty alcohol producing strain FOH33 just carries the quadruple deletion without enhancing precursor supply and fatty acid synthesis. Because we found that the fatty acid reduction (CAR activity) is limiting in fatty alcohol production the fatty acid synthesis is sufficient even without enhancing precursor supply and fatty acid synthesis (high level FFA accumulated, Fig. 5C). In future, the FFA reduction toward fatty alcohol should be further enhanced and then the fatty acid biosynthesis might need to be improved.

Question/Comment 2:

Line 27, 215, 217, & 241 Claims of "highest" or "first" should in general be deleted. The authors have no way of knowing what is or is not reported in conferences, in press, or in patents under examination.

Reply 2:

We have softened the tone according to your suggestions.

Question/Comment 3:

Line 74 Since it is already known that this pathway plays an important role in increasing acetyl-CoA for lipid accumulation in oleaginous yeasts the pathway should not be described as a novel pathway.

Reply 3:

We deleted "novel" and changed the description to "synthetic chimeric citrate lyase pathway"

Question/Comment 4:

Line 115 Please expand the explanation on why this is interesting.

Reply 4:

An explanation was added as "which may be attributed to the up-regulation of fatty acid elongation system, since yeast fatty acid synthase has much higher level production of C16 fatty acids than C18 fatty acids *in vitro*" in line 116

Responses to Reviewer 2

Question/Comment 1:

In Table 1, the authors compared cell factories for free fatty acids production, the authors could also summarize cell factories for alkanes and fatty alcohols production. The authors should introduce and cite the following previous achievements by using *E. coli* in the introduction.

Reply 1:

- 1) We added the comparison of cell factories for fatty alcohols production in Table 2. However, as alkane titer in yeast is so much lower than with *E. coli* cell factories (mentioned in line 143), we think it makes no sense to add a new table for comparing the yeast alkane production with other cell factories.
- 2) The references, about previous achievements using *E. coli*, were added

Question/Comment 2:

When single deleted 17 putative ALR/ADH encoding genes, the result indicated that in addition to ADH6, ARA1 and NRE1 could also improve the level of fatty alcohol production, so why did the authors delete these genes at the same time, or did these genes inhibit the cell growth?

Reply 2:

According to the single gene deletion results, *adh6Δ*, *gre2Δ*, *tma29Δ* *ara1Δ* and *nre1Δ* improved the specific titer of fatty alcohol. However, the improvement of *gre2Δ*, *tma29Δ* *ara1Δ* and *nre1Δ* was much less significant compared to *adh6Δ* (Supplementary Fig. 8b in revised supporting information). Furthermore, *gre2Δ* and *tma29Δ* had much less biomass yield, and *ara1Δ/nre1Δ* decreased the cell growth a bit. We thus only deleted *ADH6* for production of fatty alcohols and didn't combined the deletion of all these genes.

3. There are little mistakes in the manuscript. Examples: in line 81, " improved the can improve growth "; in Supplementary Fig. 3, "acetyl-CoA+tesA", " acetyl-CoA'tesA ", etc.

Reply 3:

We corrected these mistakes and also went through the manuscript to check other typos or mistakes.

Reviewer #1

The authors have addressed my comments at a reasonable way, therefore, I recommend to accept the paper.

Reviewer #2

Fatty acids and its derived molecules are very important molecules, and more and more researchers are focus on the overproduction of those molecules in the engineering host. Although many works have achieved a little success by using *E. coli* as the host, the baker yeast as the potential real industrial application was very difficult to engineer, the researchers need to switch the robust ethanol producer into an oil producer. In this manuscript, the authors have tried almost every metabolic engineering strategy to increase the fatty alcohol and alkane production, and this paper will be fundamental basis for the further fatty acid derived molecules overproduction in *Saccharomyces cerevisiae*. In the revised file, the authors carefully responded the reviewers' concerns one by one, and I think their answers and updates are appropriate and reasonable. I couldn't find any other useful suggestions for this and the manuscript have been polished to be ready for publication.